# Effectiveness of PROTAC BET Degraders in Combating Cisplatin Resistance in Head and Neck Cancer Cells

**DOI:** 10.3390/ijms26136185

**Published:** 2025-06-26

**Authors:** Natalie Luffman, Fereshteh Ahmadinejad, Ryan M. Finnegan, Marissa Raymond, David A. Gewirtz, Hisashi Harada

**Affiliations:** 1Department of Human and Molecular Genetics, School of Medicine, Philips Institute for Oral Health Research, School of Dentistry, Massey Comprehensive Cancer Center, Virginia Commonwealth University, Richmond, VA 23298, USA; 2Department of Human and Molecular Genetics, School of Medicine, Massey Comprehensive Cancer Center, Virginia Commonwealth University, Richmond, VA 23298, USA; 3Department of Microbiology and Immunology, School of Medicine, Massey Comprehensive Cancer Center, Virginia Commonwealth University, Richmond, VA 23298, USA; 4Department of Pharmacology and Toxicology, School of Medicine, Massey Comprehensive Cancer Center, Virginia Commonwealth University, Richmond, VA 23298, USA; 5Philips Institute for Oral Health Research, School of Dentistry, Massey Comprehensive Cancer Center, Virginia Commonwealth University, Richmond, VA 23298, USA

**Keywords:** head and neck cancer, chemoresistance, therapy-induced senescence, cisplatin, ABT-263, PROTAC, BRD4

## Abstract

Head and neck squamous cell carcinoma (HNSCC) remains challenging to treat despite multimodal therapeutic approaches. Cisplatin treatment is effective and cost-efficient, although chemoresistance and disease recurrence limit its efficacy. Understanding the mechanisms of cisplatin resistance and the identification of compounds to target resistant tumor cells are critical for improving patient outcomes. We have demonstrated that cisplatin-induced senescent HN30 HNSCC cells can be eliminated by ABT-263 (navitoclax), a BCL-2/BCL-X_L_ inhibitor that has senolytic properties. Here, we report the development of a cisplatin-resistant cell line (HN30R) for the testing of ABT-263 and the PROTAC BET degraders ARV-825 and ARV-771. ABT-263 was ineffective in sensitizing HN30R cells to cisplatin, largely due to a lack of senescence induction. However, the BET degraders in combination with cisplatin promoted apoptotic cell death in both HN30 and HN30R cells. The effectiveness of ARV-825 did not appear to depend on the cells entering into senescence, indicating that it was not acting as a conventional senolytic. ARV-825 treatment downregulated BRD4 and its downstream targets, c-Myc and Survivin, as well as decreased the expression of RAD51, a DNA repair marker. These results suggest that the BET degraders ARV-825 and ARV-771 may be effective in improving the response of chemoresistant head and neck cancer to cisplatin treatment.

## 1. Introduction

Despite all of the multimodal treatment strategies that have been developed in recent years, effective treatment of locally advanced head and neck squamous cell carcinoma (HNSCC) continues to pose a significant challenge. Although the introduction of alternative approaches such as immunotherapy with pembrolizumab/nivolumab and targeted therapy with cetuximab have improved therapeutic outcomes, cisplatin and radiation remain the most cost-efficient and effective treatment for the majority of head and neck cancer (HNC) patients [1,2]. Most HNC patients respond to cisplatin therapy initially; however, disease recurrence (local and distant relapse) leading to morbidity and mortality is all too frequent [3,4]. Chemoresistance is a complex and multilevel process that has been associated with alterations in drug import and export, the DNA repair response, and glutathione-dependent detoxification [5,6].

A key mechanism underlying chemoresistance may be therapy-induced senescence, a state of cellular dormancy caused by exposure to chemotherapeutics or radiation [7,8,9]. Therapy-induced senescence (TIS) is notably characterized by cell cycle arrest and has been associated with tumor relapse as cells leave the senescent state and recover proliferation, thus providing an attractive therapeutic target [7,8,9,10]. Senolytic drugs specifically inducing cell death in senescent cell populations are an emerging cancer therapeutic. The “two-hit” treatment regimen implementing senescence-inducing chemotherapies followed by senolytic treatment has proven to be effective in limiting proliferation and tumor growth in preclinical studies [9,11,12,13]. ABT-263 (Navitoclax) has been at the forefront of senolytics research due to its success in inhibiting BCL-X_L_ and BCL-2 in a variety of solid tumor types, including lung, melanoma, and ovarian cancer [9,11,13,14]. Our previous studies in prostate, breast, lung, and head and neck cancer demonstrated that ABT-263 acts as a senolytic, sensitizing the cells to a spectrum of antitumor modalities, including androgen deprivation (prostate cancer), doxorubicin (breast cancer), etoposide and radiation (lung cancer), and cisplatin (HNSCC) [10,11,15,16]. Other laboratories have also shown the potential impact of ABT-263 as a senolytic in breast and lung cancer [17,18,19,20]. However, ABT-263 is associated with thrombocytopenia and neutropenia, thus limiting its clinical utility and indicating the need for alternative therapeutics with fewer toxicities [12,21,22].

Recent studies have reported the effectiveness of BET inhibitors in treating both sensitive and chemoresistant solid tumors, including HNSCC, breast cancer, and colon cancer [23,24,25]. BET family proteins, including BRD2, BRD3, and BRD4, are often overexpressed in solid tumors and play a key role in activating oncogenes, stimulating proliferation, and moderating the DNA repair process, making them an attractive target for therapeutic intervention [26]. Proteolysis Targeting Chimeras (PROTACs) have emerged as an effective drug class due to their efficient degradation of target proteins via the recruitment of either Von Hippel–Lindau (VHL) or cereblon domain E3 ligases [27].

The PROTAC BET degrader ARV-825 is constructed of the OTX015 BRD4 binding motif and an E3 ligase cereblon recognition site which targets and degrades BRD4 [28]. The OTX015 warhead domain allows ARV-825 to bind and ubiquitinate BRD4, marking it for proteasome degradation [28]. As BRD4 regulates cell proliferation by targeting c-Myc and facilitates DNA repair by binding to histones on damaged chromatin, it is an attractive target for therapeutic intervention [29,30]. ARV-825 has been shown to function as a senolytic drug in hepatocellular carcinoma and colon cancer models, effectively inducing cell death in therapy-induced senescent cell populations [31]. Additional preclinical studies in neuroblastoma and gastric cancer demonstrated decreased cell viability and suppressed cell proliferation following treatment with ARV-825 [32,33]. Our laboratory has also recently shown that ARV-825 can enhance the response to estrogen deprivation therapies and prolong growth arrest in combination with standard-of-care chemotherapies in breast cancer [34,35]. ARV-771, a BET PROTAC degrader targeting BRD2, BRD3, and BRD4, has also been shown to induce cell cycle arrest and apoptosis in castration-resistant prostate cancer and hepatocellular carcinoma [36,37]. In the current work, we examined the effectiveness of ABT-263, ARV-825, and ARV-771 as senolytics in both cisplatin-sensitive and cisplatin-resistant models of HNSCC.

## 2. Results

### 2.1. Development of a Cisplatin-Resistant HNSCC Cell Line

Cisplatin continues to be the first-line chemotherapeutic agent for the treatment of HNC patients [38]. While the dose and administration schedule for cisplatin are determined according to the stage and grade of the tumor, in virtually all patients, the regimen includes multiple cycles of cisplatin infusion [38,39]. However, the development of resistance to cisplatin is not uncommon and is often responsible for the ultimate failure of therapy [6].

To develop a cisplatin-resistant cell line as a model to study the resistance mechanism, we exposed HN30 HNSCC cells to multiple cycles of 5 µM cisplatin. Sensitivity to cisplatin was determined using a standard MTS assay; the capacity of cisplatin to promote growth arrest and reduce cell viability was determined based on trypan blue exclusion, and senescence was monitored based on β-galactosidase (SA-β-gal) activity. Figure 1A indicates that IC_50_ values increased from 3.7 µM in parental HN30 cells to 10.48 µM in the HN30R resistant model. Histological and FACS-based SA-β-gal activity assays indicated that a cisplatin concentration of 5 µM, which induced approximately 80% senescence in HN30 cells, did not promote senescence in HN30R cells (Figure 1B,C). Cisplatin concentrations of 10 µM and 20 µM promoted moderate senescence (~40%) at day 5, but this senescence was reduced (<20%) by day 7 (Figure 1C). HN30 and HN30R cells maintain approximately equal doubling times, with HN30 having a doubling time of 31.1 h and HN30R taking 30.8 h. Taken together, these findings indicate that the HN30R cells are an appropriate model for studies of resistance to cisplatin treatment.

### 2.2. Cisplatin-Resistant Cells Are Not Sensitized to ABT-263

ABT-263 (navitoclax) [40,41] has been one of the most effective senolytic drugs in cancer and aging research. In our previous study, we showed the effectiveness of ABT-263 as a senolytic agent in HN30 cells that had been induced into senescence by cisplatin treatment [10].

To assess the susceptibility of the HN30R cell line to the sequential combination treatment with cisplatin followed by ABT-263, we monitored the number of HN30R cells after treatment with 5 and 10 µM cisplatin alone or in combination with 2 µM ABT-263. This concentration of ABT-263 was chosen based on the clinically achievable plasma concentration (5.7 µM) identified in a phase I trial of ABT-263 in small-cell lung cancer [21,41]. Figure 2A shows that HN30R cells treated with 5 µM cisplatin did not respond to ABT-263, although ABT-263 alone introduces a modest slowing of cell growth. Furthermore, unlike our findings in HN30 cells [10], the sequential combination of 5 and 10 µM cisplatin with ABT-263 did not significantly promote apoptosis in HN30R cells (Figure 2B).

We investigated the BCL-2 family target proteins of ABT-263 to evaluate the cisplatin resistance mechanism in HN30R. BCL-X_L_ and BCL-2 are the direct targets of ABT-263 and both contribute to the anti-apoptotic cascade. The upregulation of BCL-X_L_ and BCL-2 is associated with acquired chemoresistance as cancer cells harness this mechanism to evade apoptosis [42,43]. Additionally, BCL-2 overexpression specifically has been observed to contribute to cisplatin resistance in ovarian and lung cancer [43]. We observed that BCL-X_L_ expression does not significantly differ between HN30 and HN30R cells regardless of cisplatin treatment. However, BCL-2 expression was significantly upregulated in HN30R cells both with and without cisplatin treatment (Figure 2C). This suggests that BCL-2 expression may contribute to cisplatin resistance in HN30R cells. Furthermore, we showed that senescence is not efficiently induced with cisplatin treatment in HN30R cells compared to HN30 cells (Figure 1B,C). Together, these results suggest ABT-263 is unsuccessful in sensitizing HN30R cells to cell death due to lack of senescence mediated by cisplatin.

### 2.3. BRD4 Overexpression Decreases Overall Survival in Head and Neck Cancer Patients

BET family proteins, including BRD4, are bromodomain-containing proteins and epigenetic readers that play critical roles in transcriptional activation and various molecular pathways in cells, including tumorigenesis [44]. In a recently published study where ER-positive breast cancer cells were exposed to standard-of-care fulvestrant + palbociclib, we reported that the PROTAC BET inhibitor ARV-825 could prolong the growth arrest induced by this treatment and thereby limit or delay proliferative recovery from senescence [35]. We further indicated that a high expression of BRD4 was associated with a diminished response to standard-of-care therapy in ER-positive breast cancer [35]. Similarly, when analyzing clinical gene expression data in HNC patients, we observed a significant increase in BET family proteins. We utilized GEPIA2 bioinformatics as a resource to analyze The Cancer Genome Atlas (TCGA portal) head and neck squamous carcinoma (HNSC) dataset with a total of 563 patients. *BRD2*, *BRD3*, and *BRD4* were found to be overexpressed in HNSC tumors compared to their normal counterparts (*p* < 0.05) (Figure 3A). Additionally, overall survival analyses of patients with low versus high *BRD2*, *BRD3*, or *BRD4* expression revealed that high *BRD4* expression was associated with decreased patient overall survival through 150 months (Figure 3B). This indicates that BRD4 may have a significant role in aggressive tumor progression and may be an effective therapeutic target for combating chemoresistant HNSCC.

### 2.4. PROTAC BET Degraders Induce Senescence in the Parental and Resistant HNSCC Cells

We next examined whether targeting BRD4 with the PROTAC BET degraders can be effective in overcoming cisplatin resistance in HNSCC cells. We selected two BET degraders: ARV-825, a BRD4 preferential degrader with cereblon moiety, and ARV-771, a pan-BET degrader with VHL moiety [28,37]. Both ARV-825 and ARV-771 have been shown to effectively induce apoptosis and cell cycle arrest in solid tumors, including hepatocellular carcinoma, colon cancer, neuroblastoma, and gastric cancer [32,33,36,37]. As shown in Figure 4A, the HN30 and HN30R cells demonstrate essentially identical sensitivity for ARV-825 (IC_50_ of ~50 nM) and ARV-771 (IC_50_ of ~70 nM). Western blotting confirms that ARV-825 and ARV-771 treatments both result in BRD4 and BRD2 degradation while BRD3 shows no significant change in expression (Figure 4B).

We further monitored cell viability over time after either ARV-825 or ARV-771 exposure for 96 h via trypan blue exclusion in both HN30 and cisplatin-resistant HN30R cells (Figure 4C). Both ARV-825 and ARV-771 induced a prolonged state of growth arrest in both HN30 and HN30R cells, indicating that they may be promoting senescence. Flow cytometry staining with DDAOG, a fluorescent substrate of β-galactosidase, in HN30 and HN30R cells treated with and without ARV-825 or ARV-771 confirmed senescence induction with drug treatment (Figure 4D). Both ARV-825 and ARV-771 promoted approximately 40% senescence induction, and there was no significant difference between HN30 and HN30R in their respective levels of senescence after either ARV-825 or ARV-771 treatment. Furthermore, both HN30 and HN30R cells demonstrated moderate proliferative recovery 9–15 days after drug treatment with either ARV-825 or ARV-771. These data support the conclusion that ARV-825 and ARV-771 have senescence-inducing properties that may prolong growth stasis within the tumor cell population.

### 2.5. PROTAC BET Degraders Sensitize Parental and Resistant HNSCC Cells to Cisplatin

We next examined the effectiveness of ARV-825 and ARV-771 in combination with cisplatin in parental HN30 and cisplatin-resistant HN30R cells. Both HN30 and HN30R cells were treated with 5 μM cisplatin for 48 h, followed by either 50 nM ARV-825 or 70 nM ARV-771 for 96 h. HN30 cells treated with cisplatin alone began to recover proliferation after 9–12 days, whereas HN30R cells grew back more aggressively, recovering after only 2–4 days (Figure 5A). The combination treatment of cisplatin and ARV-825 prolonged growth arrest beyond 12 days following treatment in both cell lines (Figure 5A). Combination treatment with ARV-771 displayed a similar growth pattern as combination treatment with ARV-825, where both HN30 and HN30R cells maintained extended growth arrest 15 days after treatment (Figure 5A). However, neither HN30 or HN30R cells recovered proliferation with ARV-771 treatment following cisplatin, suggesting that ARV-771 may induce pronounced and durable cell death compared to ARV-825. At day 6, the combination treatment of cisplatin and ARV-825 induced approximately 20% apoptosis in HN30 cells and 40% apoptosis in HN30R cells (Figure 5B). In contrast, ARV-771 treatment following cisplatin resulted in approximately 60% apoptosis in both cell lines (Figure 5B). Together, these studies indicate that ARV-825 and ARV-771 both effectively sensitize parental HN30 and cisplatin-resistant HN30R cells to cisplatin treatment.

### 2.6. Suppression of BRD4 Promotes a Reduction in Target Proteins and the Induction of DNA Damage in HN30R Cells Treated with Cisplatin and ARV-825

Compared to BRD2 and BRD3, BRD4 has been found to significantly contribute to cell survival through its regulation of telomere elongation and DNA repair as well as through its downstream targeting of proliferation markers such as c-Myc [29]. We observed that both ARV-825 and ARV-771 promote downregulation of BRD2 and BRD4 (Figure 4B). Here, we focused solely on ARV-825 and its contribution to BRD4 degradation due to its significant role in cancer cell survival and proliferation. One major function attributed to BRD4 is the capacity to contribute to the DNA damage response at double-stranded DNA break sites, specifically via homologous repair machinery and its regulation of the chromatin remodeling complex SWI/SNF necessary to recruit the RAD51 DNA repair protein. Studies have shown that BRD4 binds to acetylated histones near breakpoints on DNA strands and initiates the assembly of DNA repair machinery, thus functioning as both an epigenetic reader and as a key player in DNA repair [45].

We first confirmed that 50 nM ARV-825 degraded the BRD4 protein in both HN30 and HN30R cell lines (Figure 6A). As an epigenetic regulator, BRD4 has been found to target a spectrum of genes, including cell proliferation and survival regulators such as c-Myc and Survivin, respectively [30,46]. We performed a Western blot analysis to assess the levels of c-Myc and Survivin in HN30 and HN30R cells after treatment with ARV-825. As expected, BRD4 was notably lower with ARV-825 treatment both with and without cisplatin in HN30 and HN30R (Figure 6A, Lanes 3–4, 7–8). Survivin expression was notably lower with cisplatin treatment in the HN30 cells, while it remained expressed in the HN30R cells, suggesting that the HN30 cells were sensitized to cell death with cisplatin alone (Figure 6A, Lanes 2 and 6). In addition, c-Myc and Survivin expression were significantly reduced in the ARV-825-treated HN30 and HN30R cells, both with and without cisplatin (Figure 6A, Lanes 3, 4, 7 and 8). This is consistent with the growth suppressive properties of ARV-825 alone.

BRD4 has been found to facilitate activation of the DNA repair pathway via its function as an epigenetic reader as well as through its regulation of NHEJ1, XRCC4, XRCC5, WRN, and other genes associated with DNA repair mechanisms [29]. Additionally, BRD4 closely regulates non-homologous end-joining repair (NHEJ) and has been observed to be necessary for the NHEJ repair of radiation-induced double-stranded breaks in prostate cancer [29,47]. Furthermore, previous studies in prostate cancer, triple-negative breast cancer, and cervical cancer found that the inhibition of BRD4 upregulates γH2AX and increases cell death, emphasizing the role of BRD4 in DNA repair [47,48,49].

RAD51, a DNA repair marker, was significantly downregulated with ARV-825 treatment both alone and in combination with cisplatin in HN30R (Figure 6B, Lanes 3–4). At the same time, γH2AX, a DNA damage marker, was significantly upregulated in HN30R cells treated with ARV-825 both with and without cisplatin, indicating the accumulation of damaged DNA (Figure 6B, Lanes 3–4). Taken together, this data supports the conclusion that ARV-825 may facilitate DNA damage accumulation by impairing the DNA repair pathway, thus promoting cell cycle arrest and apoptosis with cisplatin treatment.

### 2.7. ARV-825 Does Not Appear to Be Acting as a Senolytic

The result in Figure 5B prompted us to determine whether ARV-825-induced cell death was a consequence of the drug action as a senolytic (i.e., requiring senescent cells). Cisplatin-treated HN30 and HN30R cells were sorted by FACS, using our C_12_FDG sorting protocol [50], into the highest 30% and lowest 20% senescent population. We confirmed the senescence states in the sorted cells (Figure 7A), then exposed both populations to ARV-825, and monitored apoptosis using Annexin-V/PI staining. Figure 7B shows that both the senescence-high and the senescence-low populations from both cell lines exhibit virtually similar levels of sensitivity (apoptosis) to ARV-825. This result indicates that ARV-825, unlike ABT-263, does not specifically target the senescent population. This is consistent with our previous studies examining ARV-825 activity in ER-positive breast tumor, which supports the conclusion that ARV-825 indiscriminately induces cell death in both senescent and non-senescent cell populations [34]. Hence, this BRD4 degrader has a greater potential to target a broad range of tumor cells, including chemotherapy-resistant cells, not necessarily limited to senescent cells.

## 3. Discussion

Senolytic drugs have been investigated as a strategy for targeting therapy-induced senescent cancer cells and reducing the potential for proliferative recovery and tumor relapse [7]. As conventional chemotherapies for HNSCC treatment, such as cisplatin, are often dependent on inducing DNA damage, which consequently may promote senescence in a sub-population of cancer cells, senolytics could provide an effective strategy of eliminating residual senescent cells [13,51]. ABT-263 (navitoclax) has been demonstrated to function as a senolytic that sensitizes HNSCC cells to cisplatin exposure [10] and has shown similar effectiveness in breast, lung, and prostate models [9,11,15,16]. As a BH3-mimetic, ABT-263 binds with high affinity to the BCL-2 and BCL-X_L_ proteins, interfering with their interaction with pro-apoptotic BAK and BAX proteins in senescent cells, thus allowing activation of the apoptotic cascade [20].

Therapy-induced senescent cells often upregulate BCL-X_L_ as well as other anti-apoptotic proteins. Thus, ABT-263 may be an effective senolytic in sensitizing tumor cells to chemotherapies. We have previously shown that the HN30 and HN12 HNSCC cell lines undergo a temporary senescent state after cisplatin treatment, where sequential treatment with ABT-263 eliminates a significant portion of the senescent population [10]. However, we show here that cisplatin-resistant HN30R cells treated with cisplatin followed by ABT-263 did not undergo increased senescence or apoptosis, thus indicating that ABT-263 is not sensitizing the resistant cells to this form of chemotherapy (Figure 2C). This may be due to the increased expression of anti-apoptotic BCL-2 in HN30R cells compared to HN30 cells, potentially contributing to HN30R chemoresistance.

Chemoresistance represents a significant challenge to the treatment of aggressive cancer types, including head and neck cancer. Though the cisplatin chemotherapy regimen is initially successful in the early stage of the disease, it gradually becomes less effective as remaining subpopulations of tumor cells develop cisplatin resistance. As resistant cells are unresponsive to an initial dose of cisplatin and do not enter into senescence, traditional senolytics are rendered ineffective, thus requiring alternative treatments. We hypothesized that ARV-825 might act as a senolytic in HNSCC cells by functioning as a BRD4 PROTAC degrader as it was observed to induce senolysis and autophagy-dependent apoptosis via its function in non-homologous end-joining repair and autophagy gene expression [31]. Our previous studies in ER-positive breast cancer also revealed that ARV-825 treatment successfully downregulated BRD4 and c-Myc expression and prolonged growth arrest while further sensitizing breast cancer cells to palbociclib + fulvestrant treatment [34,35].

We observed that treatment with ARV-825 alone in HN30 and HN30R cells induces senescence (Figure 4D); however, the combination treatment with cisplatin resulted in a significant induction of apoptosis in both cell lines (Figure 5B). We also examined the effectiveness of a pan-BET degrader, ARV-771, in sensitizing chemoresistant HNSCC. ARV-771 induced similar levels of senescence in both HN30 and HN30R cells compared to ARV-825 (Figure 4D). However, we focused the remainder of our analyses on ARV-825 as both ARV-825 and ARV-771 downregulate BRD4 (Figure 4B), a key regulator in cancer cell survival and proliferation.

By degrading BRD4, a variety of downstream target genes are downregulated, thus suppressing the BRD4-mediated regulation of cell proliferation and survival [28,51]. We found that ARV-825 successfully degraded BRD4 and that the levels of BRD4 target proteins, Survivin and c-Myc, were significantly reduced in both HN30 and HN30R cells (Figure 6A, Lanes 3–4, 7–8). This result suggests that the suppression of BRD4 via ARV-825 may be inhibiting cell proliferation and survival through these target proteins. Furthermore, ARV-825 treatment significantly downregulates RAD51 expression, thus inhibiting its ability to sufficiently promote DNA repair. This may contribute to the accumulation of damaged DNA, as observed with increased γH2AX expression (Figure 6B, Lanes 3–4). Increased γH2AX expression is also consistent with the increased induction of apoptosis with ARV-825 treatment in HN30R (Figure 5B), together suggesting that DNA damage coupled with the inhibition of cell proliferation and survival by cisplatin and ARV-825 may force the cells into apoptosis.

These studies have collectively highlighted the utility of ARV-825 as a potential alternative therapeutic strategy for overcoming cisplatin-resistance compared to treatment with ABT-263. As previously discussed, HN30 and HN30R reflect differing levels of senescence with 5 μM cisplatin treatment, thus impacting their response to ABT-263 senolytic activity (Figure 2). In contrast, HN30 and HN30R cells both consistently express BRD4 regardless of cisplatin treatment and senescence status (Figure 6A). We found that ARV-825 treatment equally targets and degrades BRD4 and downregulates its downstream targets c-Myc and Survivin (Figure 6A) both with and without cisplatin treatment, thus sensitizing both HN30 and HN30R cells to apoptosis (Figure 5B). Interestingly, ARV-825 does not function as a conventional senolytic as high senescent cell populations induced by cisplatin did not experience a significantly different level of apoptosis compared with low senescent cell populations (Figure 7). These data support the premise that the combination of cisplatin and ARV-825 (and ARV-771) effectively induces apoptosis in both parental and cisplatin-resistant cells regardless of their senescence statuses.

BET inhibitors and PROTAC BET degraders are currently being examined in multiple clinical trials within solid tumor types and have displayed limited toxicities [24]. In vivo studies of ARV-825 and ARV-771 alone have demonstrated reduced tumor growth in addition to limited toxicities [32,36,37]. However, future in vivo studies would be needed to evaluate the impact of ARV-825 and ARV-771 in combination with cisplatin on regulating tumor growth and toxicities in chemoresistant HNSCC tumor models.

In conclusion, our data with cisplatin-sensitive HN30 and cisplatin-resistant HN30R cells suggests that ARV-825 and ARV-771 alone and in combination with cisplatin may have potential utility as an effective and alternative therapeutic strategy for HNC.

## 4. Materials and Methods

### 4.1. Cell Lines and Drug Treatment

Investigations were carried out on an HPV-negative human HNSCC cell line, HN30, which was provided by Andrew Yeudall (Augusta University, Augusta, GA, USA). The cells were cultured in DMEM (Thermo Fisher, Waltham, MA, USA) supplemented with 10% (*v*/*v*) fetal bovine serum (R&D Systems, Minneapolis, MN, USA), 100 U/mL penicillin G sodium, and 100 µg/mL streptomycin sulfate (Thermo Fisher) at 37 °C with 5% CO_2_. Cisplatin (MedchemExpress, South Brunswick, NJ, USA, HY-17394) was dissolved in water, and ABT-263 (Navitoclax) (MedchemExpress, HY-10087), ARV-825 (MedchemExpress, HY-16954), and ARV-771 (MedchemExpress, HY-100972) were dissolved in DMSO and administered at the desired concentrations.

### 4.2. Cell Viability Assay

Cell viability was determined by monitoring the number of viable cells over time using a trypan blue dye exclusion assay before, during, and after the drug treatment. The cells were collected using 0.25% trypsin/EDTA at specific time points, stained with 0.4% trypan blue (ThermoFischer, 15250061) and counted using a hemocytometer under light microscopy or TC20 Automated Cell Counter (Bio-Rad, Hercules, CA, USA, 145-0001).

### 4.3. In Vitro Drug Sensitivity Assay

The sensitivity of HN30 and HN30R cells to cisplatin/ARV-825/ARV-771 treatment was determined by MTS assay or WST-1 assay. For the MTS assay, metabolically active cells are able to catalyze tetrazolium salt 3-(4,5-Dimethylthiazol-2-yl)-5-(3-carboxymethoxyphenyl)-2-(4-sulfophenyl)-2H-tetrazolium (Abcam, Waltham, MA, USA) into a formazan product that is soluble in culture media [52]. The cells were seeded in 96-well plates and treated with different concentrations of cisplatin, ranging from 0 to 30 µM. After 24 h of exposure, the cells were treated with 10% MTS in culturing media and incubated at 37 °C for 4 h. The spectrophotometric absorbance of the samples was measured by a plate reader at the wavelengths 570 and 750 nm.

For the WST-1 assay, metabolically active cells similarly cleave tetrazolium salt into formazan via tetrazolium-reductase. The cells were seeded in a 96-well plate and treated with ARV-825, ranging from 0 to 200 nM for 96 h. The plates were then treated with WST-1 reagent (Sigma-Aldrich, St. Louis, MO, USA, 5015944001) for 4 h at 37 °C. A Promega GloMax multi-detection system plate reader was used to measure the spectrophotometric absorbance at a wavelength of 450 nm.

### 4.4. SA-β-Galactosidase Staining/Enrichment

Histochemical staining of SA-β-gal was performed as previously described [53]. Images were taken by a bright field inverted microscope (Olympus inverted microscope IX70, 20× objective, Q-Color3™ Camera; Olympus, Tokyo, Japan). The C12FDG flow cytometry was performed using the protocol described in [50]. At specific time points, the cells were collected, washed with PBS, and analyzed by flow cytometry using BD FACSCanto II and BD FACSDiva at the Virginia Commonwealth University (VCU) Flow Cytometry Core Facility (Richmond, VA, USA). To enrich the senescent population, the cells were seeded at high density for 1–2 × 106/150 mm dish and cultured overnight. The next day, the cells were treated with cisplatin and were stained with C_12_FDG on either day 5 or day 7, as indicated above. The cells were then sorted by FACS.

DDAOG (DDAO galactoside) (ThermoFischer, D6488) staining was implemented to assess senescence in ARV-825 treated HN30 and HN30R cells [54]. The cells were seeded and treated with either DMSO vehicle or ARV-825 and harvested 96 h after treatment. The cells were incubated in a 1:1000 DMEM-Bafilomycin solution at 37 °C for 30 min, followed by 1 h incubation in the dark with DDAOG at a 1:500 dilution. The cell samples were then washed three times in ice-cold 0.5% BSA. The washed cell pellets were then resuspended in 1% BSA and analyzed using a BD LSRFortessa-X20 cytometer available at the VCU Flow Cytometry Core Facility.

### 4.5. Annexin-V/PI Staining

Apoptosis quantification was performed using an Annexin V-FITC apoptosis detection kit (556547, BD Biosciences, Franklin Lakes, NJ, USA). The cells were seeded, treated with cisplatin/ABT-263/ARV-825, and harvested at the indicated time points. After washing the samples with PBS, the cells were resuspended in 100 µL of 1× Binding Buffer and incubated for 15 min in the dark at room temperature. Up to 500 µL of extra binding buffer was added to the final suspension, and then, the samples were analyzed by FACS using a BD LSRFortessa-X20 cytometer available at the VCU Flow Cytometry Core Facility.

### 4.6. Western Blotting

Western blotting was performed as described [10]. The antibodies used were BRD4 (Cell Signaling, Danvers, MA, USA, E2A7X), BRD3 (AB Clonal, Woburn, MA, USA; A2277), BRD2 (AB Clonal, Woburn, MA, USA, A16241), c-Myc (Cell Signaling, Danvers, MA, USA, D84C12), Survivin (Cell Signaling, 71G4B7), RAD51 (Calbiochem, San Diego, CA, USA, PC130; Cell Signaling, D4B10), BCL-X_L_ (Cell Signaling, 54H6), BCL-2 (Santa Cruz Biotech, Dallas, TX, USA, sc-509), phospho-histone H2A.X (Cell Signaling, 20E3) and GAPDH (Cell Signaling, D16H11) at a 1:1000 dilution, and anti-rabbit IgG HRP linked (Cell Signaling, 7074S) or anti-mouse IgG HRP linked (Cell Signaling, 7076S) at a 1:2000 dilution. ImageJ was implemented to conduct the densitometry analysis for each Wstern blot.

### 4.7. Statistical Analysis

Unless otherwise indicated, all quantitative data is shown as mean ± SEM from at least three independent experiments, all of which were conducted in triplicates or duplicates. GraphPad Prism 6.0 software was used for statistical analysis. All data was analyzed using either a one- or two-way ANOVA, as appropriate, with Tukey or Sidak post hoc, with the exception of C_12_FDG data, which was analyzed with unpaired, Student’s *t*-tests.

## Figures and Tables

**Figure 1 ijms-26-06185-f001:**
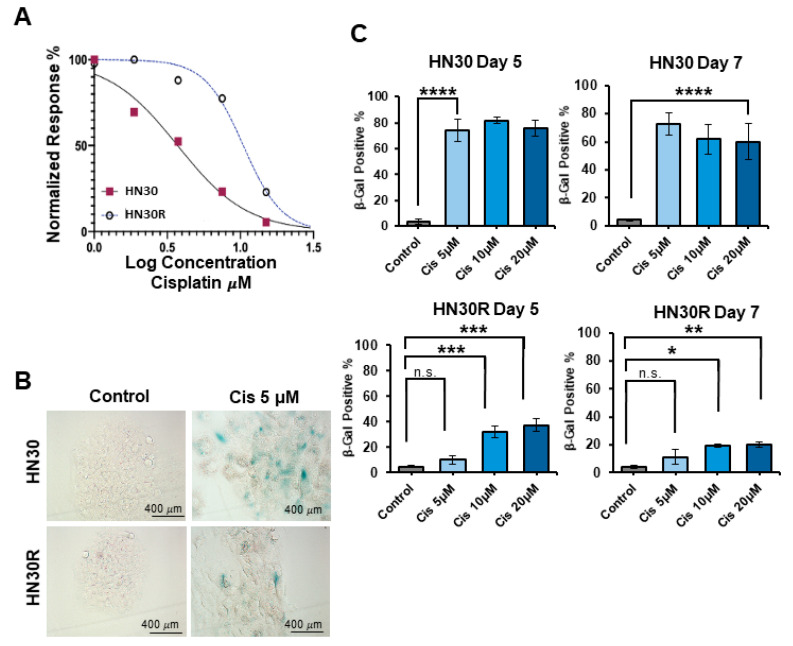
Continuous treatment with cisplatin induces acquired resistance in HN30 cells. (**A**) Parental (HN30) and cisplatin-resistant (HN30R) cell lines were treated with cisplatin at the indicated concentrations for 24 h. Cell viability at 72 h was determined using an MTS assay with three biological triplicates at each concentration. (**B**) The beta-galactosidase (SA-β-gal) activity was assessed using X-gal staining at day 5 and imaged via bright field microscopy at 20×. (**C**) C_12_FDG FACS analysis was implemented to monitor SA-β-gal activity at days 5 and 7 in both HN30 and HN30R cells following treatment with the indicated concentration of cisplatin. An averages of eight biological replicates is shown for each condition. These graphs show mean ± SEM from at least three independent experiments. * *p* < 0.05, ** *p* < 0.01, *** *p* < 0.001, **** *p* < 0.0001. n.s. = not significant.

**Figure 2 ijms-26-06185-f002:**
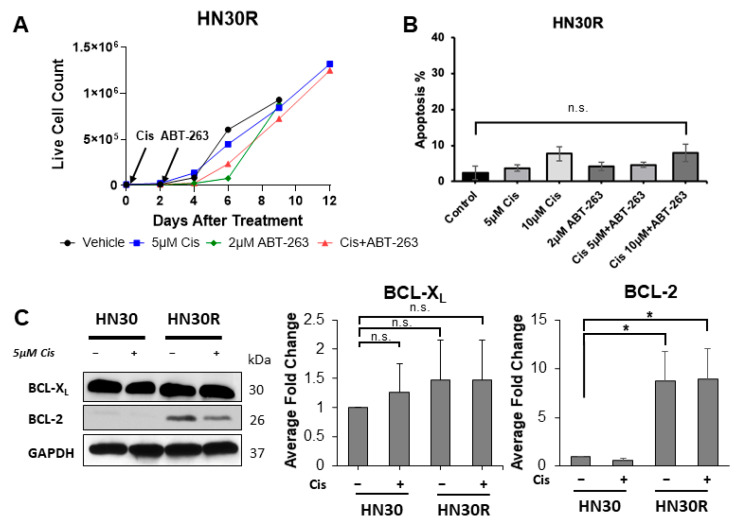
HN30R cells are not sensitized to ABT-263 by cisplatin. (**A**) HN30R cells were treated with 5 μM cisplatin for 48 h, followed by 2 μM ABT-263 for 24 h. Live cell numbers with three biological triplicates were assessed by trypan blue exclusion assay and counted via an automated cell counter. Arrows indicate when each compound was administered; data is representative of two independent experiments. (**B**) Apoptosis in HN30R cells treated with either 5 or 10 μM cisplatin ± ABT-263 for 24 h was determined using Annexin-V/PI FACS. Each condition is the average of six biological replicates. (**C**) HN30 and HN30R cells were treated with 5 μM cisplatin for 48 h and harvested for protein isolation 24 h after media change. Equal amounts of total cell lysates were subjected to Western blot analysis with the indicated antibodies; Western blot images are representative of three independent experiments. Densitometric analysis was conducted using ImageJ 1.53 quantification software. Densitometry data is presented as the average densitometry fold change of three biological replicates normalized to the HN30 vehicle control. All protein quantifications were normalized to GAPDH expression. * *p* < 0.05, n.s. = not significant.

**Figure 3 ijms-26-06185-f003:**
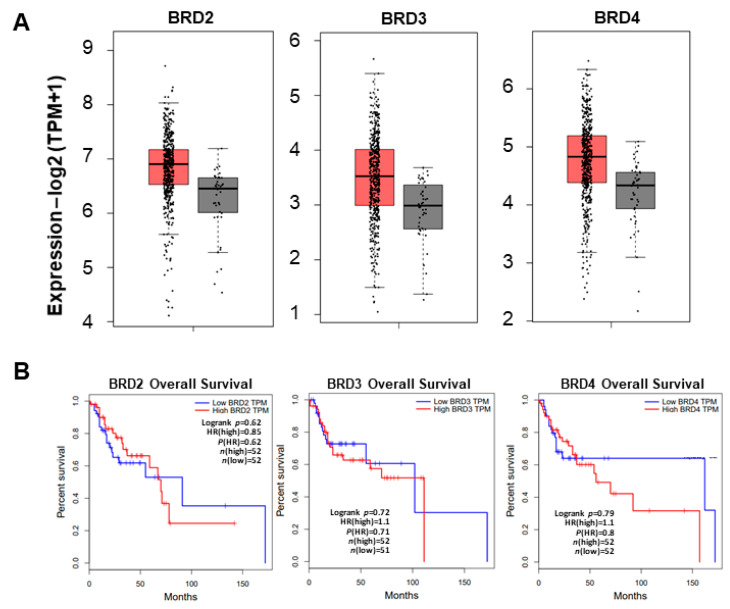
BET family mRNAs are overexpressed in head and neck tumors and contribute to worse outcomes in patients. (**A**) −log_2_ TPM + 1 mRNA expression of *BRD2*, *BRD3*, and *BRD4* in the HNSC dataset from the GEPIA and TCGA databases; *p* < 0.05. Red denotes HNSC tumor mRNA expression while grey reflects non-cancerous tissue mRNA expression. Each dot represents an individual patient sample. HNSC: head and neck squamous cell carcinoma, n_tumor_ = 519, n_normal_ = 44. (**B**) Overall survival for *BRD2*, *BRD3*, and *BRD4* low versus high TPM expression (high expression > 90%, low expression < 10%). Overall survival is decreased in patients with high BRD4 expression between 50 and 150 months of survival time.

**Figure 4 ijms-26-06185-f004:**
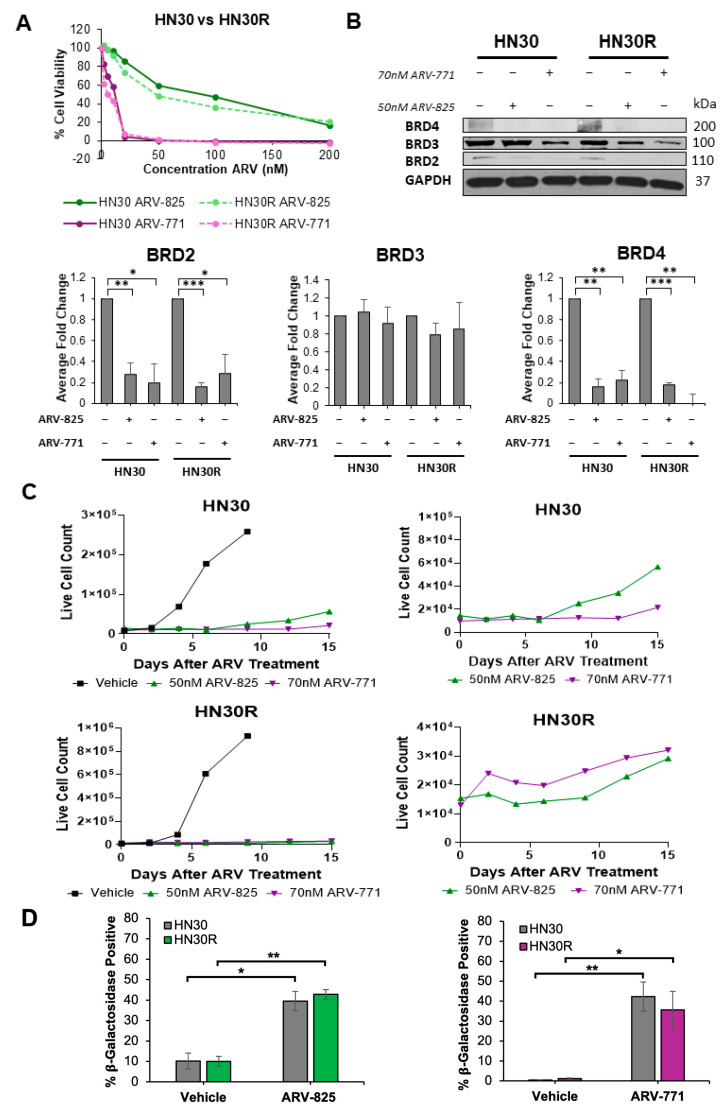
ARV-825 and ARV-771 induce senescence in parental HN30 and cisplatin-resistant HN30R cells. (**A**) HN30 and HN30R cells were treated with ARV-825 or ARV-771 at the indicated concentrations for 96 h. Cell viability was determined using a WST-1 assay with three biological triplicates at each concentration. (**B**) HN30 and HN30R cells were treated with DMSO vehicle, 50 nM ARV-825, or 70 nM ARV-771 and were harvested for cell lysates after 96 h of drug treatment. Western blot data is representative of three independent experiments and analyzed as described in Figure 2C. The levels of BRD3 do not reflect significance between any of the treatment groups. HN30 lanes 1–3 are normalized to the HN30 vehicle control (lane 1), while HN30R lanes 4–6 are normalized to HN30R vehicle control (lane 4). (**C**) At day 0, HN30 and HN30R cells were treated with DMSO vehicle, 50 nM ARV-825, or 70 nM ARV-771 for 96 h. Live cell numbers were assessed by trypan blue exclusion assay. Each timepoint is the average live cell count of three biological replicates. Enlarged graphs of HN30 and HN30R cells treated with only ARV-825 or ARV-771 are displayed in the panels on the right. (**D**) DDAOG FACS was used to quantitatively assess SA-β-gal activity for HN30 and HN30R cells treated with and without ARV-825 or ARV-771; FlowJo v10.8.1 FACS software was used to compare the vehicle and ARV-825-treated populations for each cell line. Each condition is representative of three biological replicates. * *p* < 0.05, ** *p* < 0.01, *** *p* < 0.001.

**Figure 5 ijms-26-06185-f005:**
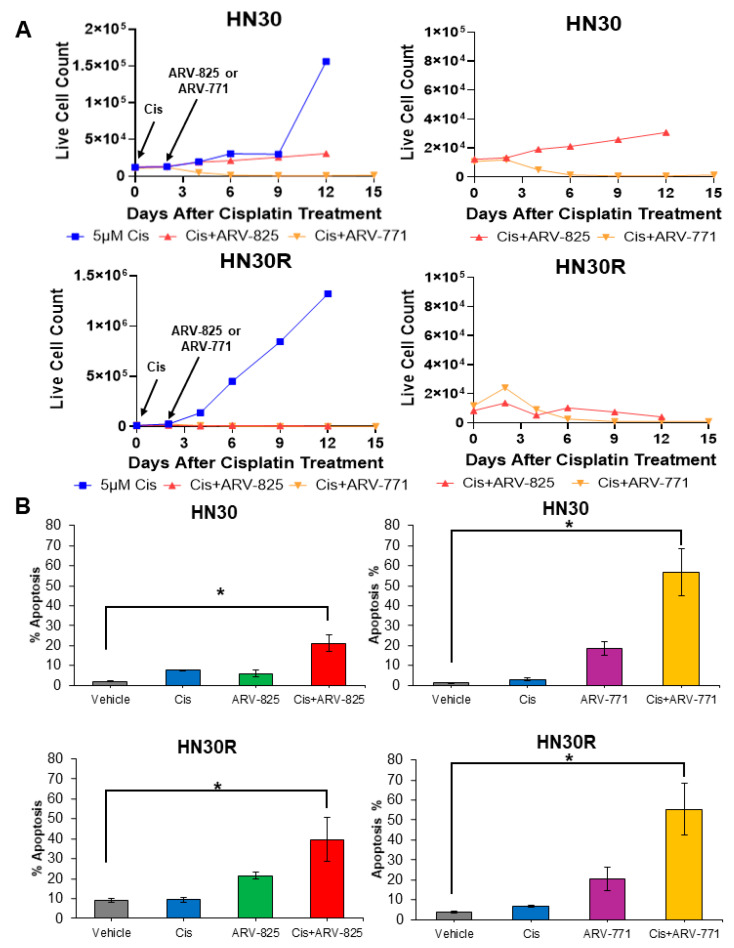
BRD4 degraders induce growth arrest and apoptosis in cisplatin-treated HN30 and HN30R cells. (**A**) HN30 and HN30R cells were treated with 5 μM cisplatin for 48 h, followed by 50 nM ARV-825 or 70 nM ARV-771 for 96 h. Live cell numbers were assessed by trypan blue exclusion assay. Arrows indicate when each compound was administered; each time-point represents the average of three biological replicates. Enlarged graphs of HN30 and HN30R cells treated with cisplatin and ARV-825 or ARV-771 are displayed in the panels on the right. (**B**) Apoptosis in HN30 and HN30R cells under identical treatment conditions at day 6 was assessed by Annexin-V/PI FACS. These graphs show mean ± SEM from at least three independent biological replicates. * *p* ≤ 0.05.

**Figure 6 ijms-26-06185-f006:**
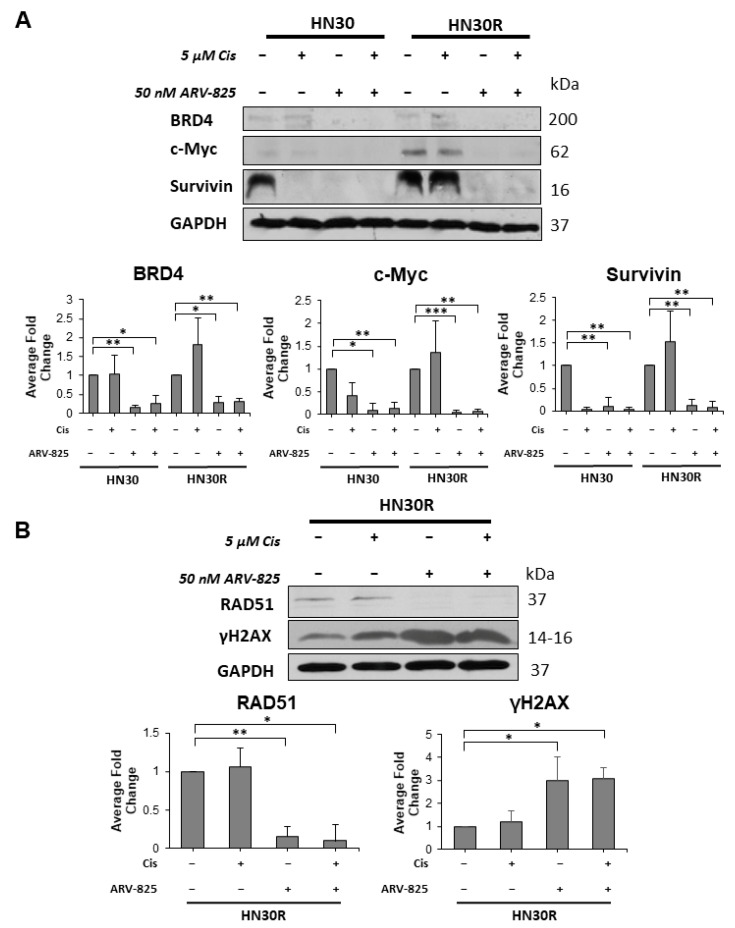
BRD4 degradation via ARV-825 treatment results in downregulation of downstream cell proliferation, survival, and DNA repair markers. (**A**) HN30 and HN30R cells were treated with 5 μM cisplatin for 48 h, followed by 50 nM ARV-825 for 96 h. Equal amounts of total cell lysates were subjected to Western blot analysis with the indicated antibodies. Densitometric analysis was conducted using ImageJ quantification software. Densitometry data is presented as the average densitometry fold change of three biological replicates normalized to the vehicle control. HN30 lanes 1–4 are normalized to the HN30 vehicle control (lane 1), while HN30R lanes 5–8 are normalized to the HN30R vehicle control (lane 5). All protein quantifications were normalized to GAPDH expression. (**B**) HN30R cells were treated with cisplatin for 48 h, followed by ARV-825 for 96 h, and harvested for Western blot analysis with the indicated antibodies. Each Western blot is representative of three independent experiments. * *p* < 0.05, ** *p* < 0.01, *** *p* < 0.001.

**Figure 7 ijms-26-06185-f007:**
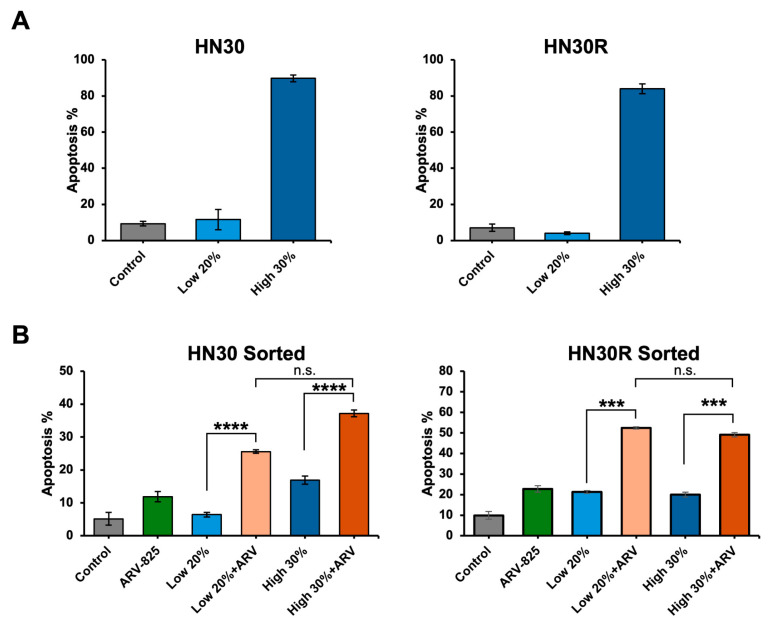
ARV-825 is not a senolytic. (**A**) HN30 and HN30R cells were treated with 5 µM cisplatin, stained with C_12_FDG, and sorted on day 3. The highest 30% and lowest 20% senescent population was analyzed via FACS. (**B**) Sorted cells were re-plated and treated with 50 nM ARV-825 for 96 h, and apoptosis was assessed using Annexin V/PI staining followed by FACS analysis. Average of each condition is representative of six biological replicates. These graphs show mean ± SEM from at least three independent experiments. *** *p* < 0.001, **** *p* < 0.0001, n.s.: not significant.

## Data Availability

All original data and images are contained within the article.

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
