# Peer review of "Effectiveness of PROTAC BET Degraders in Combating Cisplatin Resistance in Head and Neck Cancer Cells"

_ijms, 2025, doi:10.3390/ijms26136185_

Round 1
Reviewer 1 Report
Comments and Suggestions for Authors
The aim of the paper was to study the effectiveness of a PROTAC BET degrader, ARV-825, in combination with cisplatin in head and neck cancer cells. The authors did various experimental work and showed positive results. However, there are several major problems need to be addressed.
1. In line 170, the author said that BRD2, BRD3, BRD4 are significantly elevated in HNSC tumor, but according to my understanding and GEPIA analysis, these three proteins are not significant overexpressed in HNSC tumor. Similarly, the overall survival results are also not significant (Fig. 3B). Therefore, targeting BRD4 is meaningless for HNSC tumor.
2. The Western blot results in the paper lack the statistical analysis data, which is very important for readers.
3. The Fig. 1c HN30 Day 7 is marked without significance.
4. The western blot bands in Figure 7 are partially unclear, with air bubbles on some bands.
5. The results are too preliminary to support the conclusion of the paper.
Author Response
Comment 1: In line 170, the author said that BRD2, BRD3, BRD4 are significantly elevated in HNSC tumor, but according to my understanding and GEPIA analysis, these three proteins are not significant overexpressed in HNSC tumor. Similarly, the overall survival results are also not significant (Fig. 3B). Therefore, targeting BRD4 is meaningless for HNSC tumor.
Response: We must respectfully disagree with this comment by the reviewer as Figure 3A clearly indicates that BRD2, BRD3, and BRD4 all display median expression values higher in tumor tissues compared to normal tissues (Lines 172-189, Figure 3A). BRD2 has an average of 118.86 TPM in tumor samples and 86.66 TPM in normal tissues while BRD3 has an average of 10.51 TPM in tumor samples and 6.94 TPM in normal tissues (TCGA GEPIA2). In addition, BRD4 demonstrates on average 27.45 TPM in expression in tumor samples compared to 19.2 in normal tissues (TCGA GEPIA2). Furthermore, whereas the reviewer is entirely correct that BRD2 and BRD3 do not have significant impact on overall survival, patients with high BRD4 expression have a lower overall survival rate compared to those with low BRD4 expression at least through 150 months (approximately 13 years) (Lines 185-189, Figure 3B). This difference does disappear afterwards, but the advantage of low BRD4 expression is nevertheless quite dramatic.
Comment 2: The Western blot results in the paper lack the statistical analysis data, which is very important for readers.
Response: We agree with the reviewer’s comments and have now provided densitometric analyses for each western blot based on three biological replicates. The average of the densitometry ratios for the three replicates were taken for each lane and are now presented in bar graphs with statistical analyses.
Comment 3: The Fig. 1c HN30 Day 7 is marked without significance.
Response: Thank you for noting this error. This figure has been amended to reflect significance (Figure 1C).
Comment 4: The western blot bands in Figure 7 are partially unclear, with air bubbles on some bands.
Response: We agree that the quality of the western blots was suboptimal and have provided more rigorous western blots in the current Figure 6.
Comment 5: The results are too preliminary to support the conclusion of the paper.
Response: We must respectfully disagree with this comment. We clearly demonstrate that (i) both ARV-825 and ARV-771 induce cisplatin-resistant HNSCC cells into a prolonged state of senescence-mediated growth arrest via the degradation of BRD4 (Lines 210-222, Figure 4); (ii) ARV-825 or ARV-771 treatment in combination with cisplatin promote apoptosis (Lines 243-259, Figure 5); (iii) Degradation of BRD4 also results in the downregulation of RAD51, a DNA repair marker, thus promoting the accumulation of damaged DNA (increase of γH2AX) with ARV-825 treatment both with and without cisplatin (Lines 296-310, Figure 6B). Consequently, the conclusion that the PROTAC degraders ARV-825 and ARV-771 are capable of diminishing cisplatin chemoresistance in HNSCC is supported by the data presented.
Reviewer 2 Report
Comments and Suggestions for Authors
The manuscript by Luffman et al. presents a thoughtful and insightful investigation into the role of ARV-825 in overcoming cisplatin resistance in head and neck squamous cell carcinoma (HNSCC). The work highlights promising findings regarding the potential of ARV-825 in combination with cisplatin to promote apoptotic cell death in both senescent and cisplatin-resistant cells. However, some aspects of the paper require further clarification. Below are several points and questions for consideration:
1. Have you explored the efficacy of ARV-825 in additional cisplatin-resistant HNSCC cell lines beyond HN30R to strengthen the generalizability of your findings?
2. Could you provide a more detailed analysis of the mechanism by which ARV-825 induces apoptosis in both senescent and non-senescent cells?
3. Could you provide data on the effects of ARV-825 on additional downstream targets beyond c-Myc, such as PLK1 or other cell cycle regulators?
4. The discussion mentions increased DNA damage (H2AX expression) with ARV-825 treatment. Have the authors investigated whether this DNA damage is directly responsible for the increased apoptosis observed?
5. Given that ARV-825 appears to work through a different mechanism than conventional senolytics, have the authors considered comparing its efficacy to other BET inhibitors or PROTAC degraders in HNSCC models?
6. How does the combination of ARV-825 and cisplatin affect tumor growth and metastasis in xenograft or syngeneic mouse models of HNSCC?
7. What is the toxicity profile of this combination in vivo?
I believe addressing these points would significantly enhance the impact and clinical relevance of your study. We look forward to receiving your revised manuscript.
Author Response
Comment 1: Have you explored the efficacy of ARV-825 in additional cisplatin-resistant HNSCC cell lines beyond HN30R to strengthen the generalizability of your findings?
Response: This study uniquely implemented an isogenic model to reflect chemoresistance in tumor cells obtained from the same patient. The cisplatin-resistant HN30R cells were developed from the cisplatin-sensitive HN30 cell line (Lines 103-116, Figure 1). Thus, this study provides proof of concept, but did not investigate additional cisplatin-resistant HNSCC cell lines, which will be a topic for future studies.
Comment 2: Could you provide a more detailed analysis of the mechanism by which ARV-825 induces apoptosis in both senescent and non-senescent cells?
Response: The discussion section (Lines 390-416) has been modified to further explain the possible mechanism of action by which ARV-825 is sensitizing HN30 and HN30R cells.
Comment 3: Could you provide data on the effects of ARV-825 on additional downstream targets beyond c-Myc, such as PLK1 or other cell cycle regulators?
Response: Thank you for this insightful comment. We chose to evaluate BRD4’s direct target proteins, c-Myc, Survivin, and RAD51, due to their role in cell proliferation, survival and DNA damage (Lines 272-310). As PLK1 is not a direct downstream target of BRD4 and requires a mediator to interact with it, evaluating PLK1 is out of the scope of this study.
Comment 4: The discussion mentions increased DNA damage (H2AX expression) with ARV-825 treatment. Have the authors investigated whether this DNA damage is directly responsible for the increased apoptosis observed?
Response: We modified the discussion section to elaborate on how γH2Ax expression is associated with increased apoptosis with ARV-825 treatment (Lines 390-402). γH2AX expression increases with ARV-825 and is highest in combination with cisplatin in HN30R cells. This is consistent with the combination treatment displaying the highest level of apoptosis in HN30R cells (Figures 5-6). In addition to downregulation of the DNA repair protein RAD51 with ARV-825 treatment in HN30R cells, we speculated that an accumulation of damaged DNA contributes to increased apoptosis induction in HN30R (Figure 6).
Comment 5: Given that ARV-825 appears to work through a different mechanism than conventional senolytics, have the authors considered comparing its efficacy to other BET inhibitors or PROTAC degraders in HNSCC models?
Response: We appreciate this thoughtful recommendation. As recommended, we selected an additional PROTAC BET degrader, ARV-771, to evaluate in our HNSCC model. We chose ARV-771 as it is specific to VHL E3 ligases and targets BRD2 and BRD3 in addition to BRD4. As was the case with ARV-825, ARV-771 alone also induces prolonged growth arrest and senescence as well as effectively sensitizing both experimental cells lines to cisplatin (Figures 4-5). We further observed that ARV-771 induces a higher level of apoptosis compared to ARV-825 in combination with cisplatin (Figure 5B). This precise mechanism would be needed to study in the future.
Comment 6: How does the combination of ARV-825 and cisplatin affect tumor growth and metastasis in xenograft or syngeneic mouse models of HNSCC?
Response: We agree that studies in tumor bearing animal models would increase the impact of this work. Unfortunately, we are unable to perform this additional work due to lack of funding for this specific line of research.
Comment 7: What is the toxicity profile of this combination in vivo?
Response: In vivo studies of ARV-825 and ARV-771 alone have reflected reduced tumor growth in addition to no significant toxicities [1–4]. However, future in vivo studies would be needed to evaluate the impact of ARV-825 and ARV-771 in combination with cisplatin on regulating tumor growth and toxicities in chemoresistant HNSCC tumor models. Again, as indicated above, we are unable to perform this additional work due to lack of funding for this specific line of research.
References
- Li, Z.; Lim, S.L.; Tao, Y.; Li, X.; Xie, Y.; Yang, C.; Zhang, Z.; Jiang, Y.; Zhang, X.; Cao, X.; et al. PROTAC Bromodomain Inhibitor ARV-825 Displays Anti-Tumor Activity in Neuroblastoma by Repressing Expression of MYCN or c-Myc. Front. Oncol. 2020, 10, 574525, doi:10.3389/fonc.2020.574525.
- Liao, X.; Qian, X.; Zhang, Z.; Tao, Y.; Li, Z.; Zhang, Q.; Liang, H.; Li, X.; Xie, Y.; Zhuo, R.; et al. ARV-825 Demonstrates Antitumor Activity in Gastric Cancer via MYC-Targets and G2M-Checkpoint Signaling Pathways. Front. Oncol. 2021, 11, 753119, doi:10.3389/fonc.2021.753119.
- Deng, Y.; Yu, C.; Chen, L.; Zhang, X.; Lei, Q.; Liu, Q.; Cai, G.; Liu, F. ARV-771 Acts as an Inducer of Cell Cycle Arrest and Apoptosis to Suppress Hepatocellular Carcinoma Progression. Front Pharmacol 2022, 13, 858901, doi:10.3389/fphar.2022.858901.
- Raina, K.; Lu, J.; Qian, Y.; Altieri, M.; Gordon, D.; Rossi, A.M.K.; Wang, J.; Chen, X.; Dong, H.; Siu, K.; et al. PROTAC-Induced BET Protein Degradation as a Therapy for Castration-Resistant Prostate Cancer. Proc. Natl. Acad. Sci. U.S.A. 2016, 113, 7124–7129, doi:10.1073/pnas.1521738113.
Round 2
Reviewer 1 Report
Comments and Suggestions for Authors
This new version is acceptable.
Reviewer 2 Report
Comments and Suggestions for Authors
The authors have adequately addressed the previous comments and made the necessary revisions. I recommend accepting the revised manuscript.